

# Effects of auxin derivatives on phenotypic plasticity and stress tolerance in five species of the green alga *Desmodesmus* (Chlorophyceae, Chlorophyta)

Wei-Jiun Lin[1], Han-Chen Ho[2], Sheng-Chang Chu[1] and Jui-Yu Chou[1]

[1] Department of Biology, National Changhua University of Education, Changhua, Taiwan
[2] Department of Anatomy, Tzu Chi University, Hualien, Taiwan

Corresponding author
Jui-Yu Chou,
jackyjau@cc.ncue.edu.tw

## ABSTRACT

Green microalgae of the genus *Desmodesmus* are characterized by a high degree of phenotypic plasticity (i.e. colony morphology), allowing them to be truly cosmopolitan and withstand environmental fluctuations. This flexibility enables *Desmodesmus* to produce a phenotype–environment match across a range of environments broader compared to algae with more fixed phenotypes. Indoles and their derivatives are a well-known crucial class of heterocyclic compounds and are widespread in different species of plants, animals, and microorganisms. Indole-3-acetic acid (IAA) is the most common, naturally occurring plant hormone of the auxin class. IAA may behave as a signaling molecule in microorganisms, and the physiological cues of IAA may also trigger phenotypic plasticity responses in *Desmodesmus*. In this study, we demonstrated that the changes in colonial morphs (cells per coenobium) of five species of the green alga *Desmodesmus* were specific to IAA but not to the chemically more stable synthetic auxins, naphthalene-1-acetic acid and 2,4-dichlorophenoxyacetic acid. Moreover, inhibitors of auxin biosynthesis and polar auxin transport inhibited cell division. Notably, different algal species (even different intraspecific strains) exhibited phenotypic plasticity different to that correlated to IAA. Thus, the plasticity involving individual-level heterogeneity in morphological characteristics may be crucial for microalgae to adapt to changing or novel conditions, and IAA treatment potentially increases the tolerance of *Desmodesmus* algae to several stress conditions. In summary, our results provide circumstantial evidence for the hypothesized role of IAA as a diffusible signal in the communication between the microalga and microorganisms. This information is crucial for elucidation of the role of plant hormones in plankton ecology.

# INTRODUCTION

Phenotypic plasticity can be broadly defined as the capacity of a single genotype to exhibit variable phenotypes in different environments and implies that a species can conquer diverse environments. Phenotypic plasticity refers to some of the changes in an organism's behavior, morphology and physiology in response to a unique environment. A well-known

example of phenotypic plasticity is changes in multicelled structures in coenobial algae. In these algae, colonies reproduce asexually by successive divisions of the protoplast within the parent cell wall, and when progeny are released, the parent wall remains. The daughter colony may be morphologically identical to the parent, or they may exhibit remarkable phenotypic plasticity. Most studies on phenotypic plasticity in coenobial algae have been conducted considering morphological responses to an abiotic factor. *Neustupa & Hodac (2005)* demonstrated that the morphological plasticity of *Pediastrum duplex* var. *duplex* is related to the pH dynamics of freshwater lakes. *Pena-Castro et al. (2004)* also reported the phenotypic plasticity in *Scenedesmus incrassatulus* in response to heavy metal stress. However, microalgae are typically associated with other microorganisms, such as zooplankton, fungi, and bacteria. Thus, studies on phenotypic plasticity of the coenobial algae have increased in number and broadened their scope from the focus on abiotic factors to biotic ones. *Hessen & Van Donk (1993)* first indicated that the presence of the grazing pressure from water flea (*Daphnia magna*) can induce colony formation in *Scenedesmus* algae. Furthermore, Lurling and his colleague proved that the induced colony formation in the presence of herbivores is considered a strategy more efficient than constitutive defenses under variable grazing risk (*Lürling & Van Donk, 1996*; *Lürling, 2003*). *Wu et al. (2013)* further revealed that the number of cells per coenobium of *Scenedesmus* increased with the population density of *Daphnia* growth, thus indicating a grazer density–dependent response.

Auxins, which constitute a class of plant hormones, have previously been suggested to regulate physiological responses and gene expression in microorganisms (*Spaepen, Vanderleyden & Remans, 2007*). Indole-3-acetic acid (IAA) is one of the most physiologically active auxins that can be produced by numerous microbial species (*Spaepen, Vanderleyden & Remans, 2007*; *Fu et al., 2015*). Furthermore, phylogenetic analyses have revealed that IAA biosynthetic pathways evolved independently in plants, bacteria, algae, and fungi (*Fu et al., 2015*). The convergent evolution of IAA production leads to the hypothesis that natural selection might have favored IAA as a widespread physiological code in these microorganisms and their interactions. In natural water bodies, the crucial physical associations and biochemical interactions between microalgae and other microorganisms are generally well recognized (*Natrah et al., 2014*). *Piotrowska-Niczyporuk & Bajguz (2014)* found that IAA plays a crucial role in the growth and metabolism of *Chlorella vulgaris* during a 72-hour culture period. *Jusoh et al. (2015)* indicated that IAA can induce changes in oil content, fatty acid profiles, and expression of four genes responsible for fatty acid biosynthesis in *Ch. vulgaris* at early stationary growth phase. In addition, the significance of these interactions in algal phenotypic plasticity has attracted considerable scientific attention (*Lürling & Van Donk, 1996*; *Lürling & Van Donk, 2000*; *Lürling, 2003*). Furthermore, IAA has been detected in some species of Scenedesmaceae microalgae (*Mazur, Konop & Synak, 2001*; *Prieto et al., 2011*). We previously used IAA as a signal molecule in microorganisms to simulate a selection pressure caused by interspecific competition. The results indicated that the mean number of cells per particle of *Desmodesmus opoliensis* and *D. komarekii* decreased gradually as the IAA concentration increased gradually. The proportion of *Desmodesmus* unicells in monocultures increased with IAA concentration.
We also demonstrated that these unicells exhibited a lower tendency to sedimentation than did large cells and that shrinkage may facilitate nutrient uptake and light capture (*Chung et al., 2018*). However, whether other coenobial algal species of *Desmodesmus* use the same strategy to overcome stress remains unknown. Hence, the objective of the present study was to compare the effects of IAA at different concentrations on phenotypic responses in different *Desmodesmus* species. Moreover, to address the auxin specificity of these processes and obtain an insight into the complex auxin-related regulatory mechanism(s) in algal physiology, we have selected a group of compounds called "auxin analogs," such as synthetically produced naphthalene-1-acetic acid (NAA) and 2,4-dichlorophenoxyacetic acid (2,4-D), which are structurally related to IAA. We thus aim to determine the differential effects of auxins and auxin-like compounds on the morphological responses of these coenobial algae. In addition, we investigated the effects of inhibitors of auxin biosynthesis and auxin transport in *Desmodesmus*. Here, 4-biphenylboronic acid (BBo), a potent YUCCA enzyme inhibitor and an *Arabidopsis* growth inhibitor, and 2,3,5-triiodobenzoic acid (TIBA), a polar auxin transport inhibitor, were used (*Dhonukshe et al., 2008*; *Kakei et al., 2015*). To elucidate the physiological changes induced by phytohormone treatment, we also investigated whether IAA pretreatment promotes an enhanced stress-tolerant phenotype. The obtained results are crucial for elucidating the role of plant hormones in microalgal physiology.

## MATERIAL AND METHODS

### Isolation and culture of microalgae

The algal strains used here were isolated from natural water bodies in Central Taiwan. Water samples with visible microalgal population were centrifuged at 3,000 × g for 10 min at room temperature to concentrate the cells and spread onto CA agar plates (with 0.8% w/v agar) (for more details see Supplemental Information). For isolating an axenic single colony from field water samples, the streak plate method was used. The algae were cultured in CA medium. Isolated algal cells were stored at −80 °C in 15%–20% glycerol. For each experiment, the alga was cultured axenically in liquid CA medium at 125 rpm in a tube rotator and grown at 25 °C under cool white fluorescent light (approximately 46.30 µmol m$^{-2}$ s$^{-1}$) with a 14:10-h light–dark period. Each algal culture sample was observed for cellular growth rates by measuring the optical density at 680 nm. The regression equation between cell density (y × 10$^5$/mL) and OD$_{685}$ (x) was derived as $y = 162.1x + 1.3463$ ($r^2 = 99.34\%$) (*Qian et al., 2009*).

### Algae identification

The algal cells were harvested by centrifugation at 3,000 × g at 25 °C for 10 min. The genomic DNA used for analysis was isolated using AccuPrep GMO DNA Extraction Kit (Bioneer, Korea). The 18S rDNA was amplified through PCR by using the following primers: 18S forward-TTTCTGCCCTATCAACTTTCGATG and 18S reverse-TACAAAGGGCAGGGACGTAAT, which yielded a fragment of approximately 1,200 bp (*Pan et al., 2011*). The PCR conditions were as follows: initial denaturation at 96 °C for 4 min; 36 cycles of denaturation at 96 °C for 30 s, annealing at 50 °C for 30 s, and extension

at 72 °C for 1 min; and final extension at 72 °C for 6 min. The ITS1-5.8S-ITS2 rDNA was amplified using the primers ITS forward1 (ACCTAGAGGAAGGAGAAGTCGTAA) and ITS reverse1 (TTCCTCCGCTTATTGATATGC), which yielded a fragment of approximately 1200 bp (*Pan et al., 2011*). The PCR conditions were as follows: initial denaturation at 96 °C for 4 min; 36 cycles of denaturation at 96 °C for 30 s, annealing at 48 °C for 30 s, and extension at 72 °C for 1 min; and final extension at 72 °C for 6 min. DNA sequencing was performed by Tri-I Biotech, Inc. (Taipei, Taiwan). The Basic Local Alignment Search Tool was used to find regions of local similarity between sequences on the website of the National Center for Biotechnology Information (http://www.ncbi.nlm.nih.gov). The voucher specimens of algal strains used in this study are deposited at Tung-Hai Algal Lab (THAL) Culture Collection Center (http://algae.thu.edu.tw/lab/?page_id=42) of Center for Tropical Ecology and Biodiversity, Tunghai University. The detailed information of each strain is provided in Table S1. Any requests can be addressed to the corresponding author.

## Experimental design

Solutions containing different concentrations of phytohormones (IAA, NAA, and 2,4-D) and auxin-related compounds (4-biphenylboronic acid and 2,3,5-triiodobenzoic acid) were prepared to investigate their influence on the growth and morphological plasticity of *Desmodesmus* strains. The concentrations of each phytohormone and compounds used in each experiment depended on the sensitivity of each species. The initial algal density in each culture was approximately $8.24 \times 10^3$ cells mL$^{-1}$. Algae were harvested after each experiment, and the proportions of different-celled populations were calculated under an optical microscope (DMRB, Leica, Germany). The proportion of different algal populations (including unicellular; two-, four-, and eight-celled; and other colonial morphs) were calculated, and the mean numbers of cells in different morphotypes were calculated. The numbers of cells per coenobium were counted by dividing the total cell number by the number of coenobia.

## Transmission electron microscopy

All specimens were prefixed in 2.5% glutaraldehyde/0.1 M sodium cacodylate buffer (pH 7.3) containing 1% tannic acid at 4 °C overnight. After washing in 0.1 M sodium cacodylate buffer with 5% sucrose for 15 min three times, specimens were postfixed with 1% osmium tetroxide in 0.1 M sodium cacodylate buffer at 4 °C overnight. Specimens were then washed in buffer, *en bloc* stained with 2% aqueous uranyl acetate, dehydrated through a graded series of ethanol and two times with 100% acetone. Specimens were infiltrated with Spurr resin overnight and embedded in fresh Spurr resin the next day. Serial ultrathin sections of approximately 70 nm were cut with a diamond knife on a Leica Ultracut R ultramicrotome (Leica, Heerbrugg, Switzerland). Ultrathin sections were collected on Formvar-coated copper slot grids (type: GS2x1, Cat. #: G2010-Cu, Electron Microscopy Sciences) and examined with a Hitachi H-7500 transmission electron microscope (Hitachi, Tokyo, Japan) at 80 kV. Images were recorded using a 2,048 × 2,048 Macrofire monochrome CCD camera (Optronics, Goleta, CA, USA).

### Stress tests

The log-phase algal cells were treated with 300 μM IAA for 24 h. The culture samples were harvested; the cells were then washed with CA medium and resuspended in the CA medium with different treatments. The initial algal density in each culture was approximately 9.86 × 10⁶ cells mL⁻¹. For osmotic shock test, the cells were incubated in the CA medium with 0.5 M NaCl. For the effects of pH value, the culture samples were resuspended in CA medium at pH 3.0 (adjusted with HCl) or at pH 8.0 (adjusted with NaOH). For oxidative stress, the cells were exposed to hydrogen peroxide at final concentration of 5 mM. The cell suspensions subjected to the aforementioned treatments were shaken at 25 °C for 15 or 30 min. For inducing heat shock, the cells were exposed to 40 °C for 10, 15, or 20 min by immersing the cultures in a shaking water bath. For cold treatment, the cultures were exposed to 4 °C for 24 h. Fractions of viable cells of each experiment were determined by plating appropriate dilutions of the cultures on CA agar plates before and after treatments. There were six replicates of each treatment. The controls (without IAA treatment) received the same treatments used throughout the procedure.

### Statistical analysis

In the experiment of effect of auxin analogs and inhibitors of auxin biosynthesis or transport on algal growth, the statistical differences between different groups were analyzed with Kruskal-Wallis one-way analysis of variance on ranks with post hoc Dunn's test. In the experiment of plastic phenotypic changes in response to auxin analogs, the proportions of colonies with different numbers of cells and mean number of cells per coenobium were compared using a one-way analysis of variance with least significant difference post hoc test. In stress tests, the significance of differences between the groups was determined using the Mann–Whitney U test. A p of $< 0.05$ was considered statistically significant. In th abovementioned experiments, data are presented as means of three replicates $\pm$ their standard deviations (SDs).

## RESULTS

### Effect of auxin analogs and inhibitors of auxin biosynthesis or transport on algal growth and phenotypic plasticity induction

In a previous study, we performed a dose–response analysis to determine the fitness effects of IAA on the coenobial alga *D. komarekii* (*Chung et al., 2018*). The results revealed that different concentrations of IAA had different effects on the growth and morphological changes of *D. komarekii*. Thus, we concluded that *Desmodesmus* can respond to the external phytohormone IAA signal and then integrate the information to initiate physiological changes. In this study, our aim was to determine whether the physiological cues of IAA-related compounds also trigger the growth and phenotypic plasticity responses in *Desmodesmus*. With respect to *D. komarekii* growth, we examined the effects of the natural auxin IAA as well as those of the synthetic auxins NAA and 2,4-D. At 300 μM, IAA, NAA, and 2,4-D clearly inhibited growth; however, IAA caused lower inhibition than did NAA and 2,4-D (Fig. 1A). This inhibitory effect was also observed at 100 and 200 μM NAA and 2,4-D, but not in the cells treated with 100 μM IAA (Figs. 1B, 1C). These

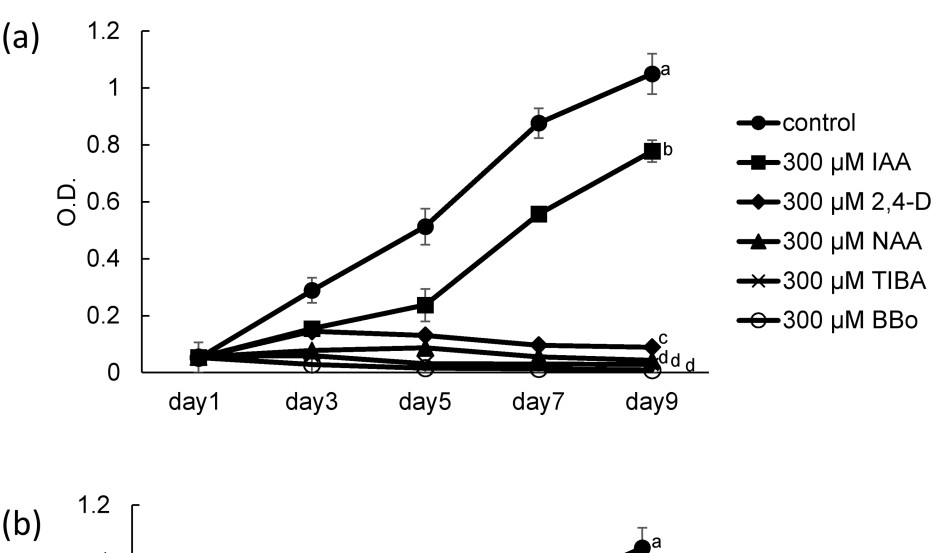

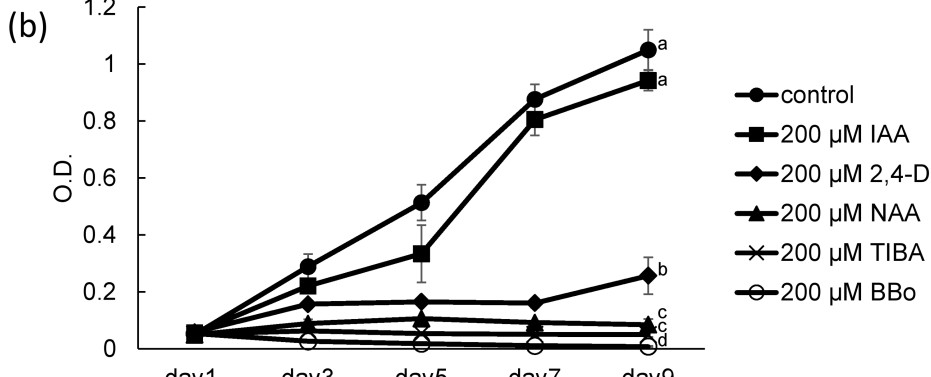

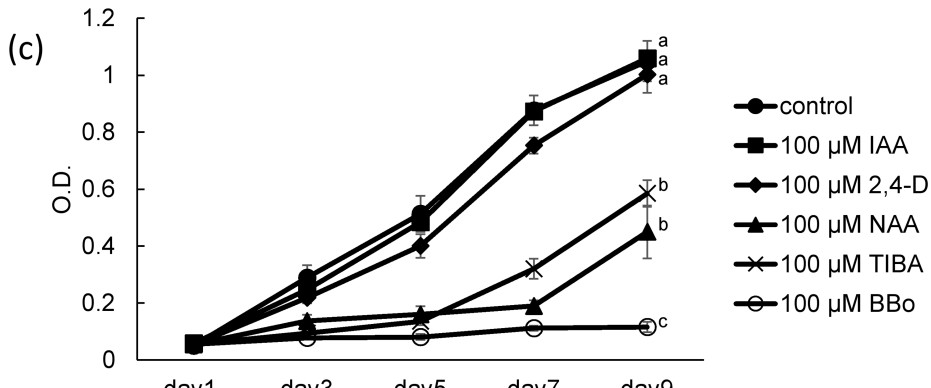

**Figure 1** **Growth of coenobial algae *Desmodesmus komarekii* in the presence of several auxins and inhibitor of auxin biosynthesis and auxin transport.** *D. komarekii* was cultured in the presence of (A) 300, (B) 200, and (C) 100 μM auxin derivatives, including indole-3-acetic acid (IAA), 2,4-dichlorophenoxyacetic acid (2,4-D), naphthalene-1-acetic acid (NAA), a polar auxin transport inhibitor, 2,3,5-triiodobenzoic acid (TIBA), or a potent YUCCA enzyme inhibitor and *Arabidopsis* growth inhibitor, 4-biphenylboronic acid (BBo). Growth curves of *D. komarekii* for each compound were measured at 1, 3, 5, 7 and 9 days. Error bars represent standard deviation of values for three replicates. Data were evaluated with Kruskal-Wallis one-way analysis of variance on ranks with post hoc Dunn's test. Different lower case letters indicate significant differences ($p < 0.05$).

observations indicated that these auxin-related compounds inhibit *D. komarekii* growth. We next tested the effects of an auxin biosynthesis inhibitor and a polar auxin transport inhibitor. BBo strongly inhibited growth even at 100 µM and its inhibitory effect increased with its concentration (Fig. 1C). At 200 and 300 µM, both of TIBA and BBo inhibited *D. komarekii* growth (Figs. 1A, 1B). These results suggested that inhibition of auxin transport and inhibition of YUCCA function both inhibit cell growth.

To measure phenotypic plasticity responses in algal populations, monocultures of *D. komarekii* were used. After 1 week of culturing, the monocultures of *D. komarekii* in the groups with exogenous 300 µM IAA and synthetic auxins were compared with those in the control environment (without treatment). We found that the monocultures of *D. komarekii* in the control groups (without IAA treatment) were dominated by one- and four-celled coenobia (Figs. 2A–2C). The morphology of *D. komarekii* monocultures changed drastically compared with the control after exposure to IAA (Figs. 2B, 2C). The proportion of unicells increased rapidly from day 3, and the proportion of four-celled coenobia decreased (Figs. 2A, 2C). The mean number of cells per particle reached its minimum on approximately day 7 (Fig. 2A), the proportion of unicells increased from 37% to approximately 73%, and the proportion of four-celled coenobia decreased from 49% to approximately 16% on day 9. The proportion of two-celled coenobia changed only slightly from approximately 13% to approximately 7%. The mean number of cells per particle in the control groups remained at >2.5 during the 9-day period. By contrast, we found that the auxin-related compounds NAA and 2,4-D both inhibited the growth of *D. komarekii* in a dose-dependent manner, but they did not influence their number of cells per coenobium/individual colony (Figs. 2D, 2E). In this experiment, the *D. komarekii* population of each culture was composed of unicells and two-, four-, and eight-celled colonies; a few three-, five-, six-, and seven-celled colonies were also present, but coenobia with more than eight cells were not observed.

Through transmission electron microscopy (TEM) analysis, we confirmed that the morphological changes in coenobia were not caused by cell aggregation but by the vegetative growth of a mother cell (Figs. 3A, 3B). No extracellular matrix was seen on or around the cells, and the connecting strands between cells were highly visible (Fig. 3C). Electron-dense kitting material (warty layer) can be seen at each corner of coenobial junction between two neighboring cells (yellow circle; Fig. 3C). Notably, we observed that specific large unicells were formed in the monocultures of *D. komarekii* after day 5 under IAA treatment. Thus, the samples collected at day 7 after IAA treatment and the cells in the control groups were used for observation of morphology through TEM. The accumulation of many starch granules and lipid bodies was observed in the large unicells compared with the cells in control groups (Fig. 3D).

The auxin-like physiological competence of selected compounds was analyzed in *Desmodesmus* based on the inhibition of growth in liquid cultures and morphological changes. Thus, we performed a dose–response analysis to determine the fitness effects of IAA and other analogs on eight other *Desmodesmus* strains. The results revealed that different concentrations of indole derivatives had divergent effects on the growth of different *Desmodesmus* species (Figs. 2 and 3, Fig. S1–S2). In general, high concentrations

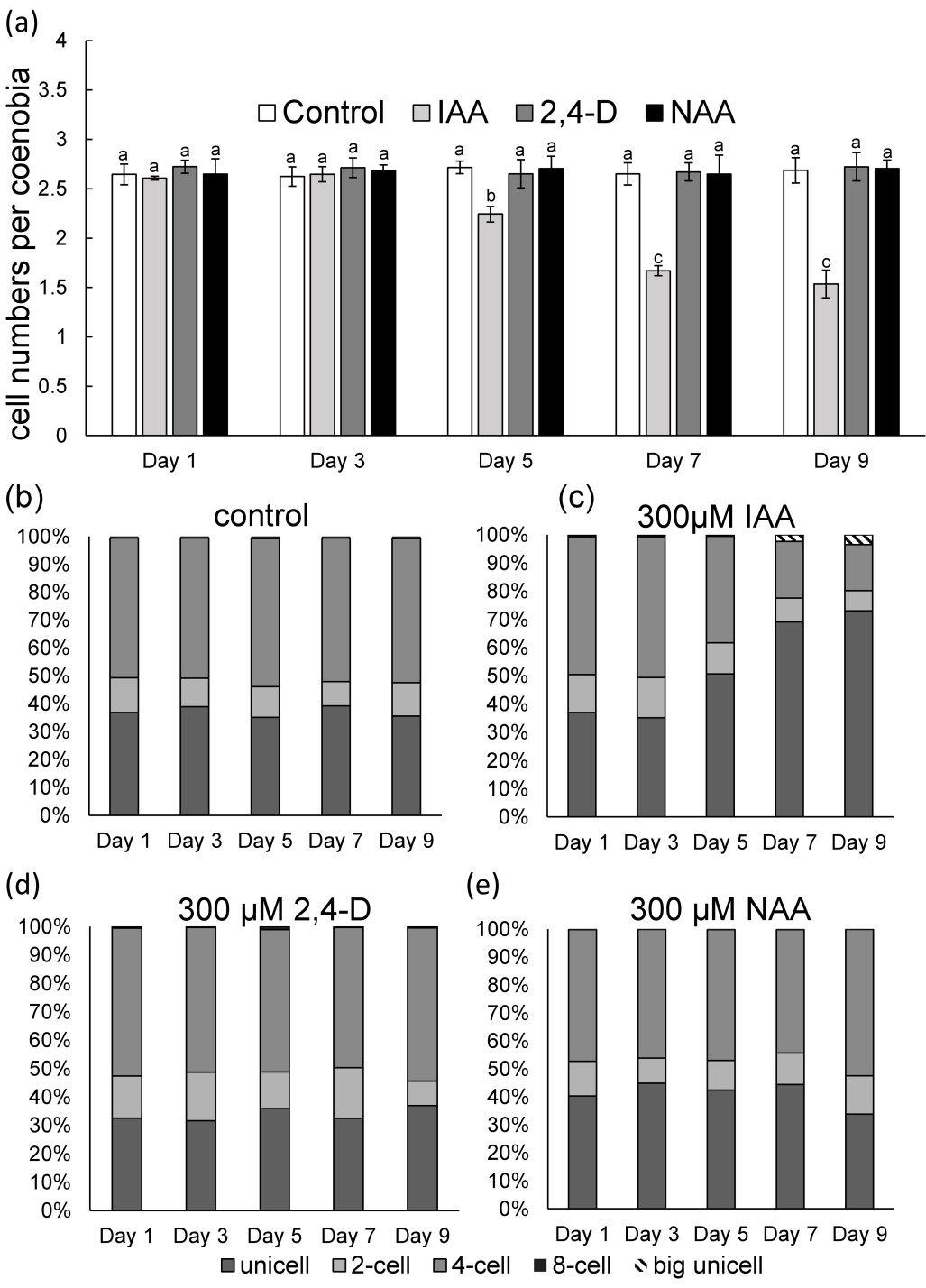

**Figure 2** (A) Mean number of cells per coenobium and (B–E) proportions of unicells and of two- and fourcelled coenobia of *Desmodesmus komarekii* cultured at 300 μM indole-3-acetic acid (IAA), 2,4-dichlorophenoxyacetic acid (2,4-D), and naphthalene-1-acetic acid (NAA) concentrations and cells without treatment. Data are presented as means (n = 3) for each group, and morphotype percentages and cell types were based on 200 cell counts in each repeat. Means with the same letter are not significantly different from each other according to a one-way analysis of variance and least significant difference post hoc test.

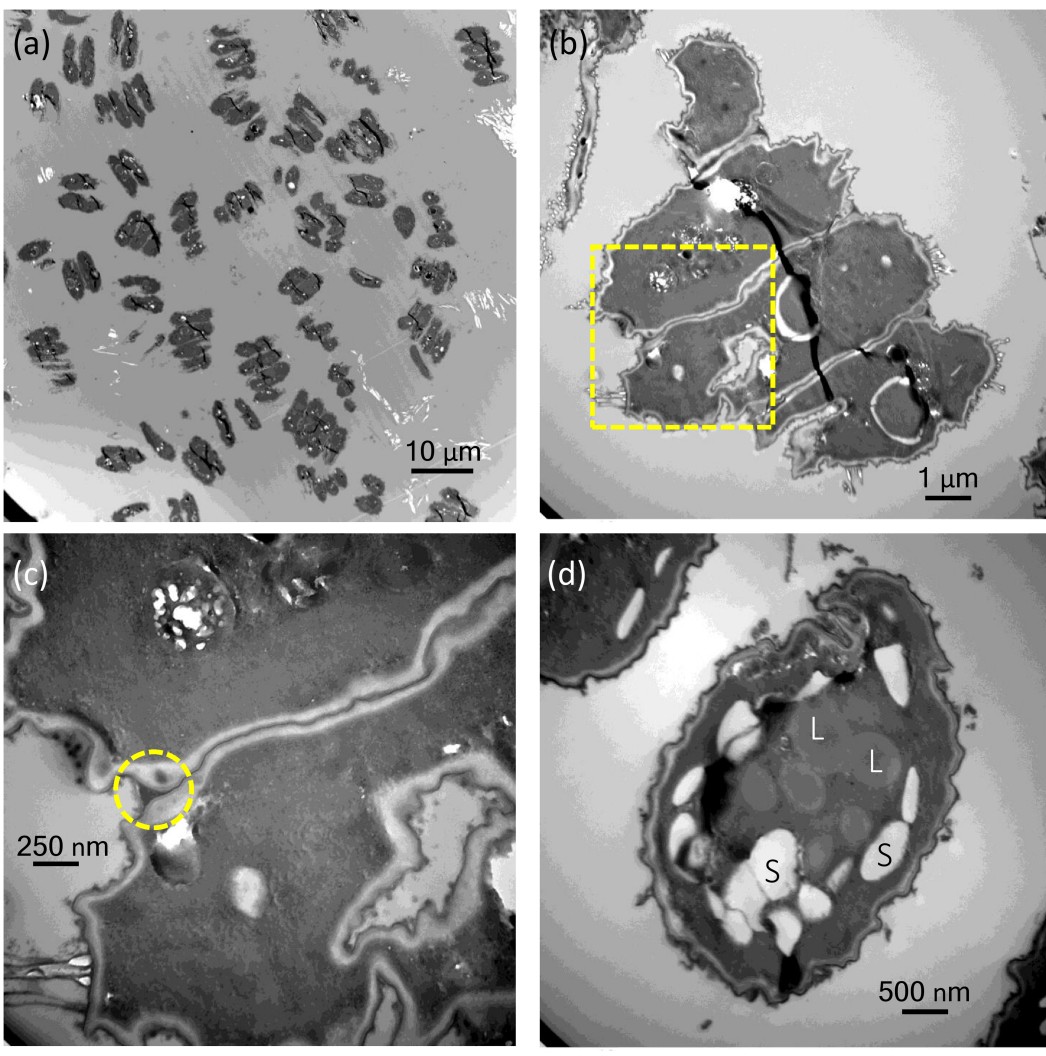

**Figure 3** **Transmission electron micrographs of *Desmodesmus komarekii* cells under indole-3-acetic acid (IAA) treatment.** (A–B) Through transmission electron microscopy, we confirmed that the morphological changes in coenobia were not caused by cell aggregation but by the vegetative growth of a mother cell. (C) No extracellular matrix was seen on or around the cells, and the connecting strands between cells were highly visible. Electron-dense kitting material (warty layer) can be seen at each corner of coenobial junction between two neighboring cells (yellow circle). (D) The accumulation of many starch granules (S) and lipid bodies (L) was observed in the large unicells at day 7 after IAA treatment compared with the cells in control groups.

(>300 µM) of IAA and other analogs inhibited the growth of the algal population. Thus, *Desmodesmus* can respond to the external phytohormone signal of IAA and other analogs and then integrate the information to initiate physiological changes. In the subsequent experiment, our aim was to determine whether the physiological cues of IAA and other analogs in these cultures also trigger phenotypic plasticity responses.
## Plastic phenotypic changes in response to IAA are strain-dependent behaviors

To measure the phenotypic plasticity responses to indole derivatives, four strains of *D. armatus*, two strains of *D. communis*, and one strain of *D. intermedius* and *D. opoliensis* were used in this study. After 1 week of treatment, the monocultures of *D. armatus* in the control groups (without indole derivative treatment) were compared with those with indole derivatives. We found that the changes in colonial morphs in *D. armatus* are specific to IAA but not to chemically synthesized auxins, NAA and 2,4-D, which are chemically more stable than IAA (Fig. 4, Fig. S1). Moreover, we found the different algal strains of *D. armatus* demonstrated phenotypic plasticity to different IAA concentrations. In *D. armatus* JYCA037, the monocultures in the control groups were dominated by two- and four-celled coenobia, with <2% unicells (Fig. 4A, Fig. S1). The morphology of *D. armatus* JYCA037 populations considerably changed under high concentration of IAA treatment compared with that in the control environment (without IAA addition). When the IAA concentration increased, the proportion of four-celled coenobia declined from >90% to approximately 21%, and the number of unicells increased from <2% to approximately 12% and two-celled coenobia increased from approximately 7% to 66%. The mean number of cells per particle of *D. armatus* JYCA037 decreased gradually as the IAA concentration gradually increased, and the cell number reached its minimum level at an IAA concentration of 400 µM (Fig. 4A). Similar results were observed in the monocultures of *D. communis* JYCA040; it was dominated by two- and four-celled coenobia in the control groups (Fig. 5A, Fig. S2). When IAA concentration increased, the proportion four-celled coenobia decreased, and the number of unicells increased. The mean number of cells per coenobium particle in the control groups of these two strains remained for >3 days after 7-day culturing. By contrast, in *D. armatus* JYCA041, the monocultures in the control groups were dominated by unicells (47%), with 47% two- and <7% four-celled individuals (Fig. 4B, Fig. S1). The morphology of *D. armatus* JYCA041 populations changed considerably under high concentration of IAA treatment compared with that in the control environment (without IAA addition). When IAA concentration increased, the proportion of two-celled coenobia increased from approximately 47% to approximately 69% and the number of unicells declined from approximately 47% to approximately 28%. The proportion of four-celled coenobia only slightly changed from approximately 6% to approximately 3%. The mean number of cells per particle of *D. armatus* JYCA041 increased gradually as the IAA concentration gradually increased and reached its maximum level at an IAA concentration of 400 M (Fig. 4B, Fig. S1) Similarly, the mean number of cells per particle of *D. armatus* JYCA039 increased gradually as the IAA concentration gradually increased and reached its maximum level when the IAA concentration was approximately 200 µM (Fig. 4D, Fig. S1). Notably, the aforementioned morphological changes were not observed in *D. armatus* JYCA045 even under treatment with high concentrations with IAA (Fig. 4C, Fig. S1). By contrast, we found that the auxin-related compounds NAA and 2,4-D both inhibit *Desmodesmus* growth in a dose-dependent manner, but the treatment did not influence their number of cells of individuals in these four *D. armatus* strains (Fig. 4). The strain-dependent response to IAA but not to NAA and 2,4-D also occurred in one strain of *D. communis* (JYCA040; Figs.

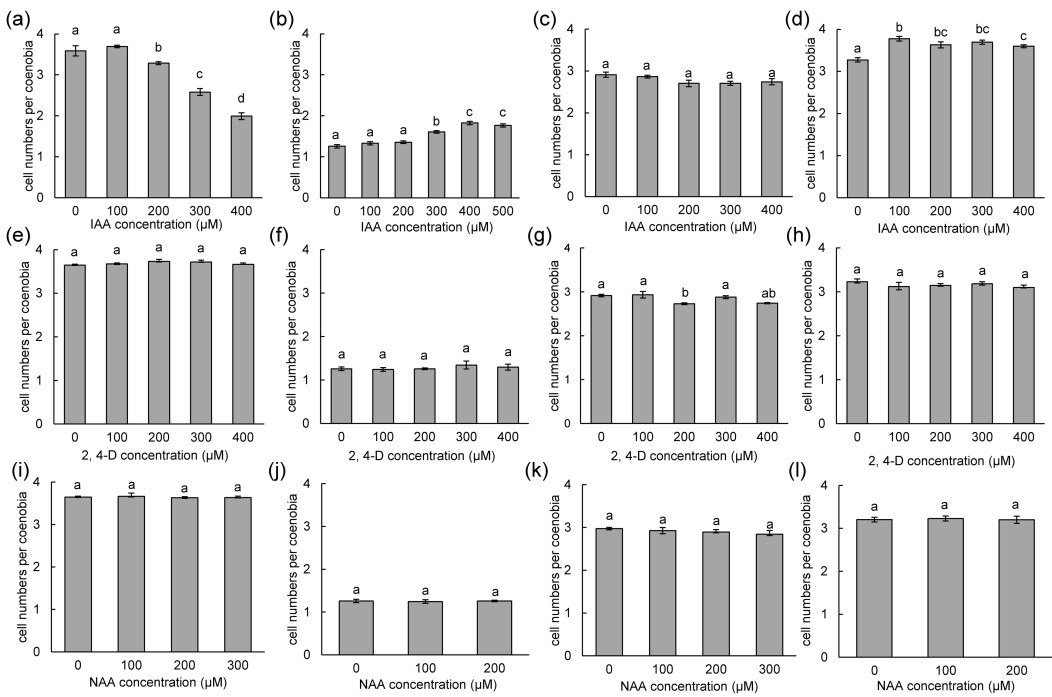

**Figure 4** **Mean number of cells per coenobium in three strains of *Desmodesmus armatus* cultured at different indole-3-acetic acid (IAA), 2,4-dichlorophenoxyacetic acid (2,4-D), and naphthalene-1-acetic acid (NAA) concentrations.** Data are presented as means ($n = 3$) for each group, and morphotype percentages and cell types were based on 200 cell counts in each repeat. Means with the same letter are not significantly different from each other according to the results of a one-way analysis of variance and least significant difference post hoc test. (A, E, I) *D. armatus* JYCA037. (B, F, J) *D. armatus* JYCA041. (C, G, K) *D. armatus* JYCA045. (D, H, L) *D. armatus* JYCA039. Data were evaluated with a one-way analysis of variance with least significant difference post hoc test. Different lower case letters indicate significant differences ($p < 0.05$).

5A, 5E, 5I, Fig. S2) and *D. opoliensis* (JYCA043; Figs. 5B, 5F, 5J, Fig. S2). However, the phenotypic plasticity caused by auxin analogs was not obviously shown in one strain of *D. communis* (JYCA044; Figs. 5C, 5G, 5K, Fig. S2) and *D. intermedius* (JYCA042; Fig. 5D, 5H, 5L, Fig. S2).

## Effect of IAA treatment on stress resistance

In this study, we found that starch granules and lipid bodies accumulated in algal cells grown at a high IAA concentration. In this environment, algal cells also demonstrated slow growth. Thus, algae contain storage in the form of natural oils, such as neutral lipids or triglycerides, and algal growth diminishes when exposed to stresses. The data reported in Table 1 showed that IAA-treated cells could withstand sudden changes in the environment, demonstrating significantly longer survival rates in the media subjected to temperature shock (40 °C, 15 min and 4 °C, 24 h), osmotic shock (0.5 M NaCl, 15 and 30 min), oxidative stress (2 mM $H_2O_2$, 30 min), acid treatment (pH 3.0, 15 min) and alkaline shock (pH 8.0, 30 min).

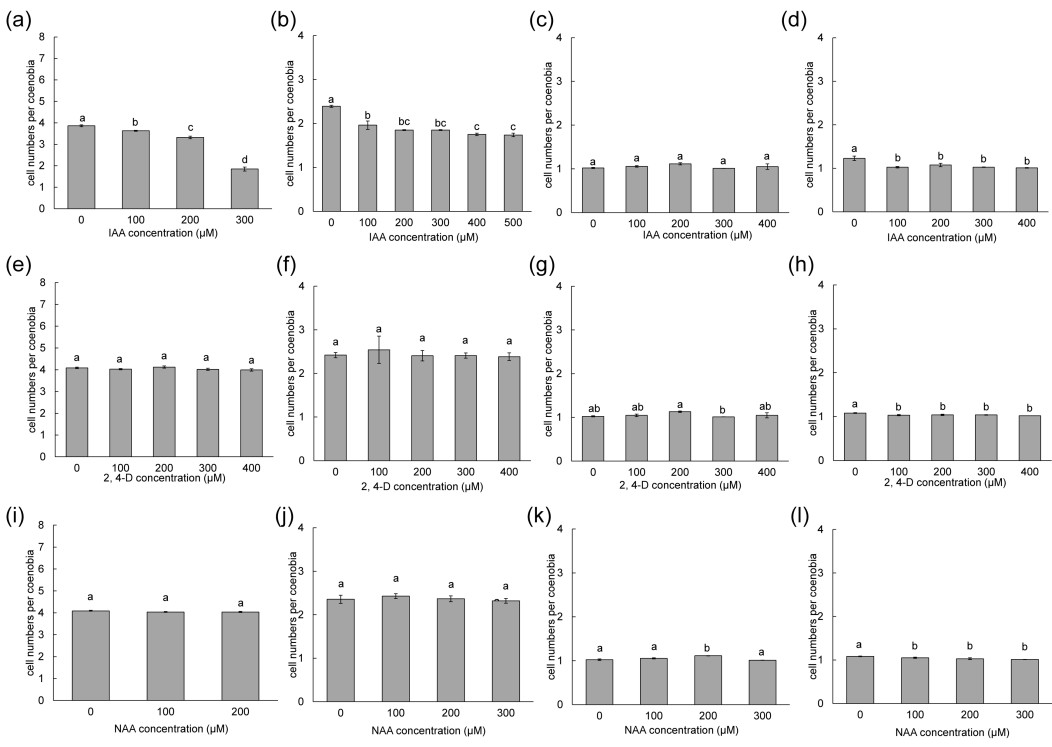

**Figure 5  Mean number of cells per coenobium of different *Desmodesmus* strains cultured at different indole-3-acetic acid (IAA), 2,4-dichlorophenoxyacetic acid (2,4-D), and naphthalene-1-acetic acid (NAA) concentrations.** Data are presented as means ($n = 3$) for each group, and morphotype percentages and cell types were based on 200 cell counts in each repeat. Means with the same letter are not significantly different from each other according to a one-way analysis of variance and least significant difference post hoc test. (A, E, I) *D. communis* JYCA040. (B, F, J) *D. opoliensis* JYCA043. (C, G, K) *D. communis* JYCA044. (D, H, L) *D. intermedius* JYCA042. Data were evaluated with a one-way analysis of variance with least significant difference post hoc test. Different lower case letters indicate significant differences ($p < 0.05$).

# DISCUSSION

A central question in biodiversity theory and ecology is the "paradox of the plankton," which indicates that the number of coexisting planktonic species far exceeds the expected and explicable number based on competition theory (*Hutchinson, 1961*). Ecologists have provided multiple solutions to the paradox by applying game theory, chaos, tradeoffs, and many other concepts in the past five decades (*Tilman, 1994*; *Huisman & Weissing, 1999*; *Károlyi et al., 2000*; *Kerr et al., 2002*; *Goyal & Maslov, 2018*). A leading theory to explain the paradox is that individual variability maintains high biodiversity in planktonic microorganisms (*Menden-Deuer & Rowlett, 2014*). In aquatic ecosystems, significant evidence supports individual variability, in individual behaviors or physiology, among planktonic microorganisms. This phenotypic plasticity has played a central role in studies on the evolution of diversity. Ecologically, phenotypic plasticity has been considered particularly crucial when environmental changes occur and different phenotypes have different fitness values across environments. This plasticity decides the survival of an individual in the face of environmental changes (*West-Eberhard, 1989*). The plasticity even

**Table 1 Increased resistance of *D. komarekii* cells to various stress conditions after exposure to IAA.**

| | Survival (%) | |
|---|---|---|
| | **Control** | **IAA-treated** |
| Heat-shock (40 °C, 10 mins) | 76.0 ± 15.4 | 87.3 ± 9.9 |
| Heat-shock (40 °C, 15 mins) | 66.6 ± 9.3 | 85.4 ± 3.5[**] |
| Heat-shock (40 °C, 20 mins) | 52.8 ± 15.8 | 62.0 ± 19.3 |
| Cold-shock (4 °C, 24 hrs) | 24.8 ± 10.7 | 50.0 ± 2.0[**] |
| Osmotic shock (0.5 M NaCl, 15 mins) | 40.7 ± 8.8 | 50.0 ± 3.6[**] |
| Osmotic shock (0.5 M NaCl, 30 mins) | 24.2 ± 12.4 | 44.5 ± 7.3[*] |
| Oxidative stress (2 mM $H_2O_2$, 15 mins) | 71.9 ± 16.4 | 73.4 ± 8.2 |
| Oxidative stress (2 mM $H_2O_2$, 30 mins) | 45.7 ± 14.7 | 65.7 ± 9.5[*] |
| Acid shock (pH 3.0, 15 mins) | 71.8 ± 7.3 | 86.3 ± 3.9[**] |
| Acid shock (pH 3.0, 30 mins) | 44.7 ± 19.1 | 61.0 ± 8.6 |
| Alkaline shock (pH 8.0, 15 mins) | 73.7 ± 14.7 | 81.4 ± 7.0 |
| Alkaline shock (pH 8.0, 30 mins) | 60.8 ± 11.7 | 80.3 ± 5.3[**] |

**Notes.**
The values reported in the table are the averages ± standard deviation of three measurements. The significance of differences between groups was determined using the Mann–Whitney *U* test.
[*] $p < 0.05$ was considered statistically significant.
[**] $p < 0.01$.

can potentiate evolvability of microorganisms by opening up new regions of the adaptive landscape (*Yi & Dean, 2016*).

In our study, we revealed that the morphological characteristics of *Desmodesmus* changed considerably when exposed to IAA compared with the algal cells in the control environment. We found that the algal strains we assayed here have different response patterns to the external IAA. In this study and our previous study, we found that when IAA concentration increased, the mean number of cells per particle of some *Desmodesmus* species decreased (*Chung et al., 2018*). The surface-to-volume ratios of the unicells was larger than the colony cells in microalgae. Changes in colony size influence algal surface-to-volume ratios, and the surface-to-volume ratio can affect light capture and nutrient uptake (*Reynolds, 2006*; *Steele, Thorpe & Turekian, 2009*). Notably, in this study, we found that in some algal strains, this trend was reversed: the mean number of cells per particle of some *Desmodesmus* strains was increased when IAA concentration increased. These colonial populations have higher sinking velocities than the unicells and two-celled coenobia; consequently, their competitive ability might be altered (*Lürling, 2003*). Thus, plasticity involving individual-level heterogeneity in behaviors and physiological characteristics is crucial for planktonic microorganisms to adapt to changing or novel conditions. This may suggest that individual variability is perhaps the key mechanism supporting planktonic biodiversity.

In this study, two widely used auxins in plant tissue culture, NAA and 2,4-D, were also used to investigate their effect on algal growth and physiological responses. These synthetic auxins show varying degrees of auxin-like activity in different bioassays (*Abebie et al., 2007*; *Savaldi-Goldstein et al., 2008*). For instance, the seedlings of *Arabidopsis thaliana* and suspension-cultured cells of *Nicotiana tabacum* BY-2 were used to investigate the
physiological activity of several auxin analogs, along with their capacity to induce auxin-dependent gene expression, to inhibit endocytosis and to be transported across the plasma membrane (*Simon et al., 2013*). The authors concluded that the major determinants for the auxin-like physiological potential of a particular compound are highly complex and involve its chemical and metabolic stability, its ability to distribute in tissues in a polar manner, and its activity toward auxin-signaling machinery. Thus, the distinct behavior of some synthetic auxin analogs suggests that they might be useful tools in investigations of the molecular mechanism of auxin action. *Ohtaka et al. (2017)* also examined the responses of the natural auxin (indole-3-butylic acid; IBA) as well as the synthetic auxins (NAA and 2,4-D) on the charophyte alga *Klebsormidium nitens*. Consistent with our results, the authors indicated that these auxin-related compounds all inhibit *K. nitens* growth in a dose-dependent manner. In their study, it was also indicated that TIBA and BBo both inhibited *K. nitens* growth. Notably, the IAA was detected in cultures of *K. nitens*, but *K. nitens* lacks the central regulators of the canonical auxin-signaling pathway found in land plants. However, the authors found that the exogenous IAA inhibited cell division and elongation, and this treatment rapidly induced expression of a LATERAL ORGAN BOUNDARIES-DOMAIN transcription factor. During evolution, *K. nitens* may have acquired a primitive auxin-response pathway to regulate transcription and cell growth. Here, we found that the natural auxin IAA and the synthetic auxins NAA and 2,4-D can all influence *Desmodesmus* growth rate. However, the changes in the colonial morphs in *Desmodesmus* are specific to IAA, but not to chemically more stable synthetic auxins. These studies have suggested that structure–activity relationships determined precisely at the level of a particular protein (e.g., receptor or carrier) may not correspond completely to the final auxin-like physiological activity of a particular compound in the streptophytes and their sister group, the chlorophytes. Thus, the comparison of the structure–activity relationships for the aforementioned phenotypic changes highlights differences in the structural requirements of these auxin-related physiological processes, thus making the differential (or the same) phenotypic outcome of the same (or different) compound a very crucial aspect of auxin biology. However, in the use of auxin-inhibitors and other experiments derived from the *Arabidopsis* methodology, we should be more careful in making conclusions, because it has not been carefully tested whether the mechanisms in *Arabidopsis* are really the same as in the green alga. In future studies, a more comprehensive examination of auxin systems of green algae, including charophyte algae, will help elucidate the origin and evolution of the plant auxin system.

Microalgae are unicellular photosynthetic microorganisms, typically found in freshwater and marine systems. The high flexibility and adaptability of this extremely diverse group of eukaryotic organisms enable it to grow in diverse environments, including fresh water, blackish, marine, and soil environments. Microalgae coexist with heterotrophic microorganisms, and the exchange of chemical compounds is central to the interactions of microalgae with other microorganisms. How microalgal–microbial interactions and participating chemical compounds shape their communities and considerably affect their fitness remains unknown (*Hom et al., 2015*). Notably, not only plants but also bacteria, fungi (including yeast), and even some microalgae produce or respond to IAA (*Fu et*

*al., 2015*). Researchers have hypothesized that the microbes sense environmental IAA concentrations to determine the cell density of its competitors (*Spaepen, Vanderleyden & Remans, 2007*; *Fu et al., 2015*; *Chung et al., 2018*). Thus, IAA has been speculated to be a signal that coordinates microbial behavior to enhance protection against damage by adverse conditions (*Bianco et al., 2006*; *Chung et al., 2018*). We proposed that the physiological changes in response to IAA confers a fitness advantage by promoting the ability of *Desmodesmus* strains to survive in their niches that often undergo fluctuations in environmental factors, such as temperature, osmotic pressure, reactive oxygen species, and pH changes. Under unfavorable stress conditions, such as nutritional starvation, salinity stress and high light intensity, lipid production is usually enhanced in algal cells, due to shifts in lipid biosynthetic pathways toward neutral lipid accumulation (*Sun et al., 2018*). Microalgae generally accumulate neutral lipids, mainly in the form of triacylglycerols (TAG) under environmental stress conditions. The accumulation of TAG likely occurs as a means of creating an energy deposit that can be readily used in response to a more favorable environment allowing for rapid growth (*Tan & Lee, 2016*). In green algae, stress conditions also trigger the accumulation of starch granules in the cells, with starch accumulation preceding the accumulation of lipid bodies following stress onset (*Siaut et al., 2011*). It is generally assumed that the starch and TAG serve as electron sinks under conditions where photosynthesis or metabolism of an exogenous carbon source remains active but the growth is limited (*Hu et al., 2008*). This phenomenon suggests that carbon sources in algal cells during stress conditions were allocated not only to storage lipid production but also starch biosynthesis, and this finding demonstrates the possibility of partitioning manipulation in the cells. To link physiological changes to phenotype, we performed various cell viability assays in response to heat, cold, osmotic stress, oxidative stress, reactive oxygen species, and pH changes. We found an increased ability to tolerate these stresses, thereby confirming the inferred enhanced stress-tolerant phenotypes when exposed to IAA. The results are consistent with earlier research on bacteria that found enhanced stress tolerance when the bacteria were pretreated with IAA across various stress conditions (*Bianco et al., 2006*; *Imperlini et al., 2009*; *Donati et al., 2013*). Here we verified that IAA increased the cell viability under many other stress conditions, but to varying extents for the different stresses. Understanding the mechanisms underlying the phenomenon, it is necessary to further investigate the effect of IAA treatment on some of the structural components of the envelope that may be involved in cellular response to stresses.

In natural water bodies, the importance of physical associations and biochemical interactions between microalgae and microorganisms is generally well appreciated, but the significance of these interactions to microbial ecology has not been investigated. In our previous study, we found that a low concentration of IAA promoted the growth of algal cells, but high concentrations of IAA inhibited cell growth (*Chung et al., 2018*). Herein, we further proved that the effects of exogenous IAA and on algal growth and phenotypic changes is species- and even strain-dependent. IAA can exert stimulatory and inhibitory effects not only on algae, fungi, and yeast but also bacteria (*Prusty, Grisafi & Fink, 2004*; *De-Bashan, Antoun & Bashan, 2008*; *Hu et al., 2010*; *Kerkar et al., 2012*; *Kulkarni et al., 2013*; *Sun et al., 2014*; *Liu, Chen & Chou, 2016*; *Fu et al., 2017*). *Bagwell et al. (2014)*

reported the frequency of co-occurrence between IAA-producing bacteria and green algae in natural and engineered ecosystems and revealed that the chlorophyll content and dry weight of algal cells were IAA concentration-dependent. A recent study also indicated that IAA produced by associated bacteria was transferred to diatoms and influenced their growth in exchange for organosulfur compounds (*Amin et al., 2015*). Thus, exposure to IAA could be likely to affect the outcome of competition among these coexisting organisms. We finally suggested that both algae and other microorganisms altered their metabolism to defend themselves form their competitors (or suit each other's needs), and this interaction is potentially very prevalent in the aquatic ecosystems. These findings indicated that IAA is a major factor determining the competition (or mutualistic interactions) between microbial species occupying the same niche. In this study, different amounts of IAA (up to 500 µM) were used to assay the effects of exogenous IAA on algal physiology. Some concerns exist that this concentration does not reflect all actual concentrations found in the wild or those produced by microbes. Many studies have shown that the amount of IAA that many bacteria and fungi can secrete was similar to or even more than 500 µM which is the highest dose we used here (*Mohite, 2013*; *Limtong et al., 2014*; *Nutaratat et al., 2015*; *Nutaratat, Monprasit & Srisuk, 2017*). According to our previous study, some fungi (e.g., *Aureobasidium pullulans*) can even can secrete more than 200 µg/mL IAA (~1,000 µM) (*Fu et al., 2016*). Furthermore, the IAA measured in this study was secreted by microbes into liquid medium. We believe that the amount of IAA produced by these microbes in the microniche under wild conditions should be higher than that we used in this study.

## CONCLUSIONS

In this study, we indicated that the changes in colonial morphs of five species of the green alga *Desmodesmus* were specific to IAA but not to the chemically more stable synthetic auxins, naphthalene-1-acetic acid and 2,4-dichlorophenoxyacetic acid. Our results also proved that IAA treatment potentially increases the tolerance of *Desmodesmus* algae to several stress conditions. Our results suggest that IAA could be used as a diffusible signal to elicit interspecific communication among different organisms. Furthermore, the plasticity involving individual-level heterogeneity in behaviors and physiological characteristics is crucial for planktonic microorganisms to adapt to changing or novel conditions.

## ACKNOWLEDGEMENTS

We thank the members of the Chou Laboratory for their helpful discussions and comments on the manuscript. We are grateful to the staff of the Electron Microscopy Laboratory, Tzu Chi University, for technical support. This manuscript was edited by Wallace Academic Editing.

## Funding
This work was supported by grants from the Ministry of Science and Technology (MOST 105-2311-B-018-001-MY3 and MOST 108-2621-B-018-002-MY3 to Jui-Yu Chou). The funders had no role in study design, data collection and analysis, decision to publish, or preparation of the manuscript.

## Grant Disclosures
The following grant information was disclosed by the authors:
Ministry of Science and Technology: MOST 105-2311-B-018-001-MY3, MOST 108-2621-B-018-002-MY3.

## Competing Interests
The authors declare there are no competing interests.

## Author Contributions
- Wei-Jiun Lin conceived and designed the experiments, performed the experiments, analyzed the data, prepared figures and/or tables, authored or reviewed drafts of the paper, and approved the final draft.
- Han-Chen Ho performed the experiments, prepared figures and/or tables, and approved the final draft.
- Sheng-Chang Chu performed the experiments, analyzed the data, prepared figures and/or tables, and approved the final draft.
- Jui-Yu Chou conceived and designed the experiments, analyzed the data, prepared figures and/or tables, authored or reviewed drafts of the paper, and approved the final draft.

## Data Availability
The raw measurements are available in the Supplementary Files.

The algal samples are available at Tung-Hai Algal Lab (THAL) Culture Collection Center (http://algae.thu.edu.tw/lab/?page_id=42) of Center for Tropical Ecology and Biodiversity, Tunghai University: THAL106 to THAL114.

## Supplemental Information
Supplemental information for this article can be found online at http://dx.doi.org/10.7717/peerj.8623#supplemental-information.

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
