# Peer review of "Effects of auxin derivatives on phenotypic plasticity and stress tolerance in five species of the green alga Desmodesmus (Chlorophyceae, Chlorophyta)"

_PeerJ, doi:10.7717/peerj.8623_

## Round 0.1 · original submission · Major Revisions

The manuscript content was viewed positively by both reviewers. There is novel and important content. However, some key points need to be addressed before the manuscript can be published.

Please carefully consider and address the comments of each of the Reviewers. In addition:

1) Title/abstract. The title uses the term “phenotypic plasticity” without explanation. This could mean any number of things. What is measured is probably better described as “colony morphology”. Neither the title nor the abstract makes it clear that we are dealing with multiple species and multiple clones of the genus. It is important that these be mentioned and it is also important that it be noted that the cultures were recent isolates; this is a positive aspect of the work since it argues that we are preserving real features of environmental variability versus those acquired in long-term culture. Some readers will be unfamiliar with auxins, so it will be important to specify in the abstract that IAA is an auxin, at first mention. The sentence “Notably, different algal species…exhibited phenotypic plasticity different to that of IAA” tells us little or nothing. I think what is meant is that other auxins had different effects on colony morphology than did IAA.

2) The Introduction is concise. In line 40 there is an unnecessary “the” in front of “coenobial algae”, in line 48 the preposition “on” should be “of”, “Furthermore” is repeated twice in a short space (lines 58 and 60), and Josoh et al. requires a year in line 68. It’s also not clear what “density of Daphnia growth” (line 54) means….simply ‘concentration of Daphnia’?

3) Materials and Methods. Details of the isolation locations of the algal strains used must be given (line 96), and consider also Reviewer 1’s comments. This can be provided as a Supplementary Table. The results can’t be reproduced or verified otherwise. Has CA medium been previously published? If so, cite the publication and omit the details here. If not, provide the details as a Supplementary Table and cite it. This will be much more useful and legible.

4) Results. First and foremost, we need to see the results of the algal identifications (see Reviewer 1 comments). Desmodesmus identifications, in general, are based on 18S and ITS (Hegewald, E., Wolf, M., Keller, A. Friedl, T. & Krienitz, L. 2010. ITS2 sequence-structure phylogeny in the Scenedesmaceae with special reference to Coelastrum (Chlorophyta, Chlorophyceae), including the new genera Comasiella and Pectinodesmus. Phycologia 49: 325-335), which the authors have determined, but we are shown no sequence data, nor referred to any desposited sequences (this is irregular). This is unacceptable. We need to be able to evaluate this in order to judge whether what we are looking at really constitute species or strains. These data need to be presented in the paper and referred to. The conclusions (species/strains) need to be explained and rationalized.
-Figures are cited out of order (2/3 before 1).
-Growth rate comparisons in Figure 1 (lines 219-222) need to be supported by statistical comparisons.
-There are far too many Figures presented for the information conveyed. A reader will get lost. This needs to be rationalized considerably. I suggest the following. ‘cell numbers per coenobia’ and the ‘percentage composition’ are simply different presentations of the same data. I recommend using ‘cell numbers per coenobia’ for Figures (they are the only data which have statistical support). The composition data could be included in Supplemental Material…in fact, it could be very easily and efficiently organized in a single Table in an Excel spreadsheet for the convenience of readers.
Next, each species could have its own Figure with three panels, IAA, 2-4D and NAA. Combine data for strains together, using multiple bars on a single graph (as in Figure 2 for time). Thus, Figures 4-7 consolidate to one three-panel Figure, Figures 8 and 10 to one three panel Figure. This makes the manuscript manageable and a reader can follow the arguments more easily.
-There are issues with the way data are handled re: IAA effects on stress responses. First, in the results, the title in line 277 “IAA Improves Algal Defenses Against Stress” is surely a conclusion that must be reserved for the Discussion. The same applies to the statement “Thus, we propose…” (lines 281-284). Relocate these to the Discussion. The arguments regarding “marginally” significant results and “not…significant” but “higher” results (lines 287-290) must be removed. One performs statistical tests at a chosen level and accepts the results. They are significant or not. If not, then one must accept that there are no differences; in the discussion, it is possible to discuss the limitations of the experimental design and how improvement might be made that could improve resolution. Inspection of Table 1 also suggests that in several cases the data are not homoscedastic (i.e. variance between treatments is not equal). If so, then a t-test is not appropriate. I would suggest that this be tested. It may well be that with proper statistical treatment (i.e. data transformation or application of non-parametric tests), different results might be found. In any case, it is troubling that the results are so inconsistent: i.e. a heat-shock of 40 °C for 15 min but not 10 or 20 min yields a difference…an acid shock at pH 3.0 of 15 min but not 30 min yields a difference. These inconsistencies weaken the conclusion substantially and should be discussed. I suggest that the current conclusions (e.g. lines 369-372) are over-stated.

·

Basic reporting

The structure and language are clear and have publication quality, except for minor corrections indicated below and very minor ones directly in the annotated PDF.

General linguistic matters
A. I strongly recommend improving the title since a genus as a merely taxonomic unit cannot have stress tolerance or something like that, but only its species or strains. Although the title would become longer, it is suggested to change the end of the title into ….in five species of the green alga Desmodesmus.
B. For the citations in the Introduction Section, I recommend only providing the citation in brackets at the end of the sentence. The present form of “XX says that…(XX + year) is redundant and puts the main message into the dependent clause and the less important information about the author into the main clause. In the Discussion Section this style appears to be ok for stressing the conclusion of the cited authors which is discussed in the light of the new findings. But here the year should be mentioned directly behind the author name: XX (year) says that… This format may be more troublesome for the Editorial Office, but given the high publication costs the Editorial Office should provide every possible service to the authors!
C. The references are ok in my random checks, but all scientific names should be given in italics!
D. In Fig. 1, writing the concentrations directly as 100, 200, and 300 “uM”(!) should be deleted, since this information is given in the bar as well as the legend, and the writing error of u for micro can be avoided.

Experimental design

The effect of plant hormones on plankton responses is investigated exemplarily experimentally with IAA and derivates for the plant hormones and different species and strains of the fresh water green alga Desmodesmus. The responses of the algae include total increase of cell numbers and numbers of cells per colony and by this also size of cells and colonies. The number of cells is important in potential interaction with other aquatic organisms. The sizes of cells and colonies additionally influences buoyancy of planktonic organisms.

The present study is one of the few experimental contributions in this field and provides important clues for interpreting field observation of plankton ecology.

I am sure that the authors have applied the correct methodology, but they need to explain several methods not described in the text as I suggested in the validity box.

The above comments are also sent to the Editor.

Validity of the findings

I am convinced that the experimental design and conclusions are all valid, but the Materials & Methods Section requires some fundamental complementation and some conclusions should be discussed more carefully:
1. Deposit of vouchers: Physical voucher strains or specimens are a basic requirement for reproducibility of the identification; lack of such vouchers in responsible journals leads to rejection. It is a particular strength of this study that the authors isolated and identified their algae themselves instead of simply ordering well-identified, but perhaps physiologically degenerated strains from a collection. In contrast to some other publications dealing with identified species, here species were identified. These new and well characterized biological resources are too important for not depositing them in a publicly accessible collection.
2. The methods for identification well described in the Materials & Methods are not mirrored in a paragraph in the Results. Please, provide the results of your species identification explicitly by briefly listing the species and numbers of strains per species. Please specify the range of bp identity in the BLAST search and the also the gap of how many bp which are different to the next closely related species.
3. The methods for measuring “growth” are not explained, but can only be concluded from Fig. 1 where OD is indicated. Please, explain how OD was used in the Materials & Methods and whether cell density is linearly correlated with the number of living cells. If chlorophyll content was used for measuring the OD, it is possible that chlorophyll content is changed under different conditions without parallel change of the cell numbers?
4. Was or how was the initial amount of algae defined for the experiments?
5. How were the concentrations of IAA and other compounds selected for the experiments? Are these concentrations similar to those in plants or in the freshwater environment? This should also be mentioned in the Discussion Section for indicating the ecological significance of the experimental findings.
6. In the use of auxin-inhibitors and other experiments derived from the Arabidopsis methodology, please, be more careful in making conclusions, if it is not been carefully tested whether the mechanisms in Arabidopsis are really the same as in the green alga. The presence of a pyrenoid is a clear evidence that physiology of algae is not just the same as in angiosperms.
7. What was the agar concentrations in the agar plates?
8. Was the CA medium autoclaved? (some reviewers/journals even require providing the brand name of each compound! I do not think that this would be important for this experiment)
9. What kind of grids and coating were used for electron microscopy?

Reviewer 2 ·

Basic reporting

In this manuscript, the effect of indole-3-acetic acid (IAA), an auxin constituting a class of plant hormones, on the phenotypic plasticity of different Desmodesmus species was investigated based on the previous findings on Desmodesmus opoliensis and Desmodesmus komarekii. The results were found to be strain-dependent in other Desmodesmus species, namely, D. armatus, D. communis, D. apoliensis and D. intermedius and discussed in terms of the advantages of individual-level heterogeneity among planktonic microorganisms. Exposure of D. komarekii to IAA lead to an increased resistance to various stress conditions similar to earlier findings on bacteria. In addition to IAA, structurally related auxin analogs, naphtalene-1-acetic acid (NAA) and 2,4-dichlorophenoxyacetic acid (2,4-D), and auxin-related inhibitor compounds (4-biphenylboronic acid and 2,3,5-triiodobenzoic acid) were also used and they were shown to inhibit the growth of Desmodesmus.
4-biphenylboronic acid (BBo) and 2,3,5-triiodobenzoic acid (TIBA) are not included in the final paragraph of introduction describing the general experimental procedure. The fact that they are types of auxin inhibitors is mentioned first in Fig 1 legend and is not understood well from the legend. They are explained later in results line 216. If an explaination is given in the introduction final paragraph, it would be easier to understand. Also the figure legend should be more explanatory.
Results starts with the title Effect of auxin analogs… However, inhibitors of auxin biosynthesis or transport was also given in this section. Therefore, it would be good to change the name of the section accordingly. Line 220 states the inhibition of D. komarekii growth with TIBA and BBo and later it is speculated that auxin transport and inhibition of YUCCA function could be the reason. It could be discussed in more detail in the Discussion section.
In the results part, line 227 says indole derivatives had divergent effects... and refers to Fig 4-11. Later in the next paragraph, Fig 4-6 are mentioned (line 240) and Figure 4 results are described later (line 241). Then it is jumped to Fig 8 (line 252). It makes confusion and breaks the coherence. I think it would be better to arrange the figure numbering in a more clear way.
Figures 4, 5, and 6 gives the results of different compounds (Fig 4 IAA, Fig 5 2,4-D and Fig 6 NAA) on three strains of D. armatus. However, Figure 7 gives the result of three compounds on the fourth strain. Why giving Figure 7 with a different version than 4, 5, and 6? Figure 7 could be deleted and the data can be presented to be a part of Fig 4, 5, and 6 as the fourth strain.


The manuscript might benefit from English editing. Several grammar errors are:
line 59 a numerous, erase “a”,
line 73 a selection pressures,
line 92 role “of” plant hormones,
line 120 instead of provided: yielded,
line 121 instead of are: were
Line 215-215 an auxin biosynthesis inhibitor”s“ erase s
Line 235 …derivatives, four strains…
Line 235 four strain”s”
Line 236 erase each
Line 239 chemically synthesized?
Line 240-241 Sentece starting with moreover needs revision
Line 572 Figure 1 legend first sentence needs revision such as: …several auxins and ihibitors of …
Line 302-305 Please restate the sentence.
Line 318-319 repeat of the same phrase
Line 359 Instead of “these”, Microalgae
Line 403 erase responses



Minor editing errors:
line 111: 3000xg,
line 118: ITS1-5.8S-ITS2,
line 124: Inc”.”
Line 344: lateral organ boundaries-domain harboring transcription factor?

References: Please go through the references to make them uniform
- Some are written long (Bianco et al., 2006), some as abbreviation (Chung et al., 2018), some with capital letter in each word, some not…
- De-Bashan et al, 2008 was written as upper case
- Microorganism names should be written in italic, a couple of examples: Donati et al., 2013, Bianco et al., 2006, … etc.

Experimental design

Original primary research within the scope of the journal, research question is well defined and relevant. How the research fills an identified gap is indicated.

Other comments are indicated in part 1

Validity of the findings

Comments are indicated in part 1.

---

## Round 0.2 · Minor Revisions

The manuscript is much improved and most of the issues have been resolved. Each of the reviewers have identified a few outstanding corrections required. Please consider and address these.

I will raise a couple additional issues to address.

1) I'm not completely satisfied with the clarification of "phenotypic plasticity", but how about these steps to help the reader:

a) in line 14 of the abstract, insert "(i.e. colony morphology") after "phenotypic plasticity". This is, after all, your principle measure of phenotypic plasticity and will make this perfectly clear to a reader.
b) then in line 22, insert "(cells per coenobium)" after "changes in colonial morphs", which will make it clear exactly what you quantified.
There's also a space missing between "plankton" and "ecology" in line 32.

2) To aid in referencing Supplementary Materials, please number the Table as Supplementary Table 1, and in the text specifically refer to the graphical materials by Supplementary Figure and Supplementary Table numbers.

·

Basic reporting

All my previous suggestions have been satisfactorily considered in the revised version. I only made some corrections of minor writing errors.

Experimental design

All my previous questions concerning methodology have been satisfactorily answered in the revised version.

Validity of the findings

The authors have clarified all the open questions addressed in my previous review.

Additional comments

Please, check a few writing errors. You did a great job with this study!

Reviewer 2 ·

Basic reporting

The manuscript is a revised manuscript and I see that the authors have improved the manuscript during revision. The coherence is better and the figures are easier to follow upon revision, but I still have some suggestions:

- Please take the sentence (line 231) “By contrast, we found that the auxin-related compounds NAA and 2,4-D both inhibited the growth of D. komarekii in a dose-dependent manner, but they did not influence their number of cells per coenobium/individual colony (Fig. 2d, e).” to the previous paragraph where you are explaining Figure 2.
- Please take the sentences (line 234-238) to the first paragraph of Results where you are explaining Figure 1.
- Please provide legends for supplementary figure 1 and 2.
- Figure 2 legend: please refer to a, b, c, d, and e parts of the figure for clarity
- Figure 5: I think D. armatus JYCA039 (part of Figure 5) should be transferred to Figure 4, because Figure 4 has the results of the other three D. armatus strains.

Minor points:
- Abstract, first sentence: algae
- Introduction, line 92: instead of Therefore, In addition.
- Line 94: an Arabidopsis…
- Line 98: elucidating
- Line 107: please transfer “for more details see Supplementary Materials” to line 105 after “with 0.8% w/v agar”
- Materials and Methods, line 187: In the abovementioned experiments
- Results, line 191, title: the inserted phrase, make the first letters capital
- Dicussion, line 358: please erase one of the a’s and use small letters
- Discussion, line 371: instead of if, because?
- Discussion, line 433: amounts…were
- Discussion, line 436: IAA that many… can secrete
- Discussion, line 436: similar to…
- Discussion, line 438: can even secrete
- Discussion, line 439: ug->μg
- Figure legends, line 619: inhibitors
- Figure legends, line 622:and a potent
- Please add Table S1 to the title of Supplementary table.

Experimental design

No additional comment

Validity of the findings

No additional comment

---

## Round 0.3 · accepted · Accept

Nicely done; this takes care of all the outstanding concerns.